# Observation of novel topological states in hyperbolic lattices

Weixuan Zhang [1,3], Hao Yuan[1,3], Na Sun[1], Houjun Sun [2] & Xiangdong Zhang [1✉]

The discovery of novel topological states has served as a major branch in physics and material sciences. To date, most of the established topological states have been employed in Euclidean systems. Recently, the experimental realization of the hyperbolic lattice, which is the regular tessellation in non-Euclidean space with a constant negative curvature, has attracted much attention. Here, we demonstrate both in theory and experiment that exotic topological states can exist in engineered hyperbolic lattices with unique properties compared to their Euclidean counterparts. Based on the extended Haldane model, the boundary-dominated first-order Chern edge state with a nontrivial real-space Chern number is achieved. Furthermore, we show that the fractal-like midgap higher-order zero modes appear in deformed hyperbolic lattices, and the number of zero modes increases exponentially with the lattice size. These novel topological states are observed in designed hyperbolic circuit networks by measuring site-resolved impedance responses and dynamics of voltage packets. Our findings suggest a useful platform to study topological phases beyond Euclidean space, and may have potential applications in the field of high-efficient topological devices, such as topological lasers, with enhanced edge responses.

[1] Key Laboratory of advanced optoelectronic quantum architecture and measurements of Ministry of Education, Beijing Key Laboratory of Nanophotonics & Ultrafine Optoelectronic Systems, School of Physics, Beijing Institute of Technology, 100081 Beijing, China. [2] Beijing Key Laboratory of Millimeter wave and Terahertz Techniques, School of Information and Electronics, Beijing Institute of Technology, Beijing 100081, China. [3] These authors contributed equally: Weixuan Zhang, Hao Yuan. ✉email: zhangxd@bit.edu.cn

Exploring novel topological phases of matter is one of the most fascinating research areas in physics[1-6]. Since the pioneering discovery of the integer quantum Hall effect in 1980[7], a large number of fascinating quantum phases with distinct topological properties have been successively proposed. These novel topological states have been revealed in various systems possessing completely different characteristics, ranging from lower dimensions to higher dimensions[8-10], from Hermitian systems to non-Hermitian systems[11-13], from periodic structures to disordered structures[14], from single-particle systems to many-particle systems[15,16], from linear lattices to nonlinear lattices[17-19], from static systems to dynamic systems[20,21], and so on. To date, most of the established topological states of matter have been principally employed in Euclidean geometry with a zero curvature.

On the other hand, the non-Euclidean geometry exists widely in nature and plays important roles in many different fields, including mathematics, the holographic principle, the general theory of relativity and so on. To experimentally explore the novel physics of curved spaces, the controllable laboratory setups are required to be constructed. Recently, using circuit quantum electrodynamics, the experimental realization of discrete hyperbolic lattices[22], which are regular tessellations in the curved space with a constant negative curvature, has stimulated many advances in non-Euclidean geometry and hyperbolic physics, including the Bloch band theory of hyperbolic lattices[23,24], the crystallography of hyperbolic lattices[25], quantum field theories in continuous negatively curved spaces[26], the hyperbolic drum in circuit networks[27] and so on[28-33]. Additionally, it is worthwhile to note that boundary sites always occupy a finite portion of the total site regardless of the size for the hyperbolic lattice due to the negative curvature. This is completely contrary to the case of Euclidean lattices, where the ratio between the number of boundary sites to that of total sites approaches to zero in the thermodynamic limit. Recently, the hyperbolic topological state has been theoretically proposed based on a tree-like design of the Landau gauge in periodic and open systems[28,32]. Inspired by these fascinating phenomena revealed in hyperbolic lattices, it is important to ask whether there are other undiscovered topological states in hyperbolic lattices, and how to construct the hyperbolic topological phases in experiments.

In this work, we report the experimental observation of two kinds of topological states, that are boundary-dominated first-order Chern edge states and fractal-like higher-order zero modes, in engineered hyperbolic lattices. In particular, by extending the original Haldane model in Euclidean space to hyperbolic lattices, unidirectional edge states with nontrivial real-space Chern numbers are proposed. We note that the Haldane model allows for the more direct (or Euclidean-like) assignment of the gauge field and Berry curvature compared to the tree-like design of the Landau gauge[32], but it is difficult to be realized in high-frequency regimes (such as using photonics) due to the requirement of next nearest neighbor couplings. Hence, in experiments, the suitably designed circuit network, where the long-range site coupling is easily to be realized, is used to construct the hyperbolic Haldane model. The impedance and voltage measurements demonstrate the key features expected of a Chern insulator, including localized edge states within a bulk gap, the chiral edge propagation, and the protection against backscattering. Moreover, based on the deformed hyperbolic lattice with unequal coupling strengths in different layers, the fractal-like midgap higher-order zero modes are revealed, and observed in the designed hyperbolic circuit network. Our finding unfolds the intriguing properties of hyperbolic topological states, and suggests a route to design highly compact topological devices with the efficient spatial utilization.

## Results

**Boundary-dominated first-order topological states in hyperbolic Chern insulators.** We start by briefly introducing the projection scheme of a hyperbolic plane with a uniform negative curvature in the $(2 + 1)$-dimensional Minkowski space onto a complex unit disk. As illustrated in Fig. 1a, under the stereographic projection with the reference point located at $(x = 0, y = 0, t = -1)$, a hyperboloid defined by $t^2 - x^2 - y^2 = 1$ could be mapped to a unit disk at $t = 0$, where the geodesics on the hyperboloid (green lines) are projected to circular arcs perpendicular to boundaries of the disk. Such a unit disk is called the Poincaré disk equipped with the hyperbolic metric. Based on this projection scheme, the hyperbolic lattice, which is a discrete tessellation of the two-dimensional hyperbolic space, could be mapped to the Poincaré disk.

To illustrate the hyperbolic lattice in the Poincaré disk, we introduce the Schläfli notation {$p$, $q$}, which represents a tessellation of the plane by $p$-sided regular polygons with the coordination number $q$, to label the lattice pattern. We note that only the triangular lattice {3, 6}, square lattice {4, 4}, and honeycomb lattice {6, 3} could exist in the two-dimensional (2D) Euclidean space, where the relationship of $(p-2)(q-2) = 4$ must be satisfied. In contrast, the hyperbolic tessellation is ensured by $(p-2)(q-2) > 4$ so that there are infinite kinds of lattice models in the hyperbolic space[7]. Here, we focus on the hexagonal hyperbolic lattice {6, 4} embedded into the Poincaré disk, as presented in Fig. 1b, where all neighboring lattice sites possess equal hyperbolic distances. More details about the geometrical properties and mathematical representations of the hyperbolic lattice are provided in Supplementary Note 1.

To construct topological states in the hyperbolic lattice, we extend the Haldane model[34] originally defined in Euclidean space to the hyperbolic lattice {6, 4}. It is noted that the property of hyperbolic tight-binding lattice model depends on the connection of all vertices, and is regardless of the configuration of vertices. Hence, the hyperbolic lattice could also be illustrated by arranging the vertices in the form of quasi-concentric rings, and maintaining the connection of all vertices unchanged. In this case, the finite hyperbolic lattice {6, 4} with a sixfold rotation invariance in Poincaré disk (shown in Fig. 1b) is equivalent to the successive quasi-concentric rings with $L = 4$ layers, as shown in Fig. 1c. For clarity, we mark lattice sites in the first, second, third and fourth layers by cyan, blue, green and red dots, respectively. By introducing nearest-neighbor (NN) hoppings ($\gamma$) and direction-dependent next-nearest-neighbor (NNN) hoppings ($\lambda e^{i\varphi}$) in each hexagon, the hyperbolic Haldane model is achieved. Detailed coupling patterns in hexagons composed of lattice sites from different layers are illustrated in right insets of Fig. 1c, where solid lines and dashed arrow lines correspond to NN hoppings and NNN hoppings, respectively. In this case, the hyperbolic Haldane model can be effectively described by a tight-binding Hamiltonian as:

$$H = \sum_{<i,j>} \gamma a_i^\dagger a_j + \sum_{<<i,j>>} \lambda e^{i\varphi} a_i^\dagger a_j + h.c. \quad (1)$$

with $a_i^\dagger (a_i)$ being the creation (annihilation) operator at site $i$. The bracket $<\ldots>$ ($<<\ldots>>$) indicates that the summation is restricted within NN (NNN) sites. $\varphi$ is the geometrical phase of NNN couplings. Compared with the Haldane model defined in Euclidean space, where each bulk site possesses three NN couplings and six NNN couplings, there are four NN couplings and eight NNN couplings for each bulk site of the hyperbolic Haldane model.

Firstly, we perform a direct diagonalization of the Hamiltonian for the finite hyperbolic Haldane lattice with $\gamma = 1$, $\lambda = 0.2$, $\varphi = 2\pi/3$ and $L = 4$. Figure 1d shows the calculated

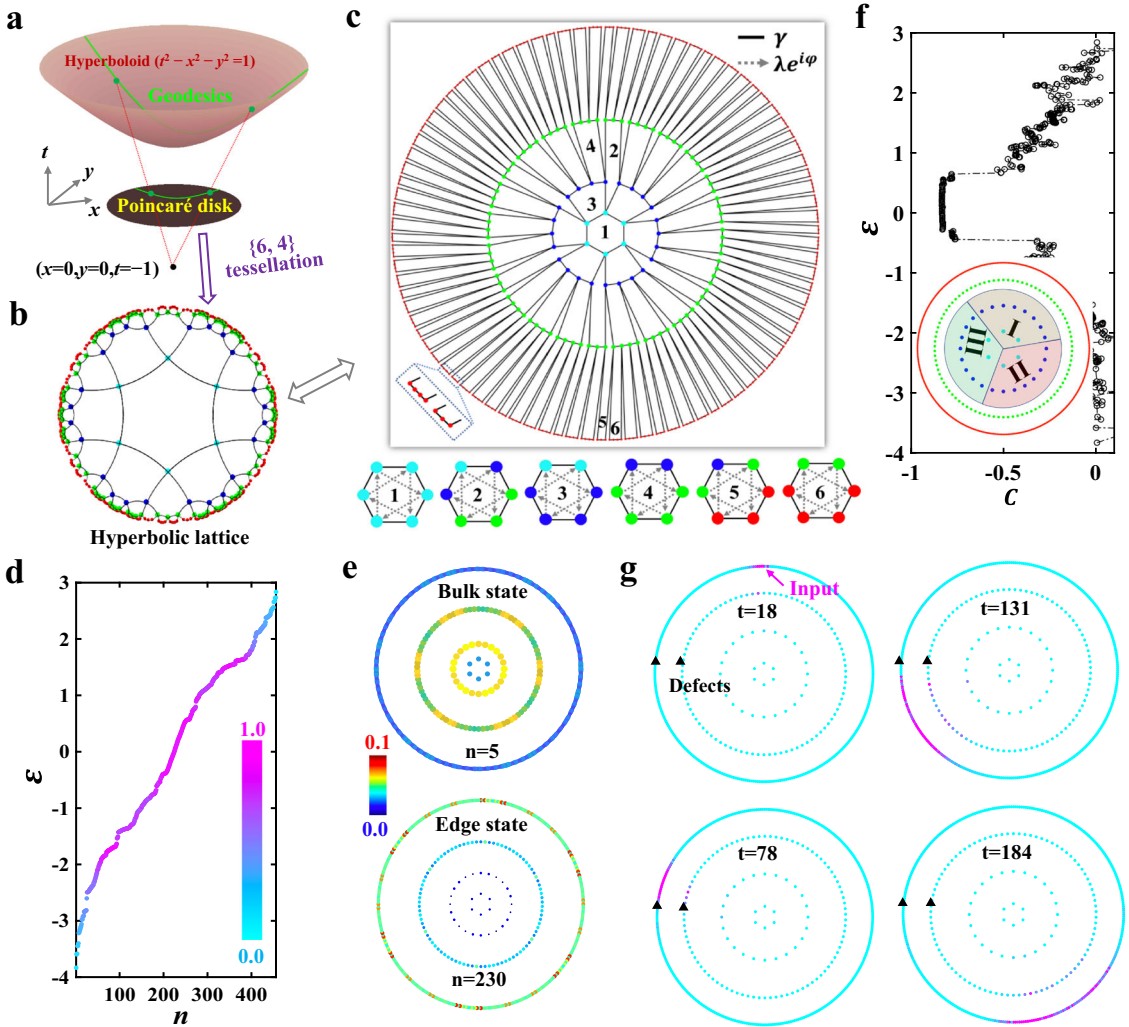

**Fig. 1 Hyperbolic Haldane model and boundary-dominated first-order topological edge states. a** The left chart illustrates the stereographic projection scheme of a hyperbolic plane ($t^2 - x^2 - y^2 = 1$) onto the Poincaré disk at $t = 0$. Green lines represent the geodesics on the hyperboloid, which are projected to circular arcs perpendicular to boundaries of the Poincaré disk. **b** The hyperbolic lattice {6, 4} embedded into the Poincaré disk. **c** The finite hyperbolic lattice {6, 4} in the form of successive quasi-concentric rings with $L = 4$ layers. The left-bottom inset displays the enlarged view of the outermost layer. Right insets plot coupling patterns in hexagons composed of lattice sites from different layers. **d** The calculated eigen-spectrum of the system with $L = 4$. The colormap corresponds to the quantity $V(\varepsilon)$ for the localization degree at the boundary. **e** Profiles of bulk and edge states with eigen-energies of $-3.445$ and $0.1406$. **f** The calculated real-space Chern number of each eigenmode. The inset plots three different regions I, II and III used in the calculation. **g** Four charts present the spatial distributions of $|\varphi_i(t)|$ at different times with $t = 18, 78, 131$, and $184$, respectively.

energy-spectrum with the corresponding eigenmode being marked by n. To quantify the localization degree of each eigenmode on the boundary (the outermost layer), a quantity $V(\varepsilon) = \sum\limits_{i \in L=4} |\phi_i(\varepsilon)|^2 / \sum\limits_{i \in L=1,2,3,4} |\phi_i(\varepsilon)|^2$ of each eigenmode is calculated, where $\phi_i(\varepsilon)$ is the complex amplitude at site $i$ with the eigen-energy being $\varepsilon$. The colormap in Fig. 1d represents the quantity $V(\varepsilon)$. It is noted that the value of $V(\varepsilon)$ approaches to 1 for the edge-concentrated eigenstate, while bulk-localized eigenmodes have a near-zero value of $V(\varepsilon)$. We find that edge states mainly locate around the zero-energy region, and bulk modes exist at the low- and high-energy ranges. To further illustrate the distribution of associated eigenmodes, in Fig. 1e, we plot spatial profiles of bulk and edge eigenmodes with energies being $\varepsilon = -3.445$ ($n = 5$) and $n = 230$. It is clearly shown that the eigenmode at $\varepsilon = 0.1406$ possesses the feature with a significant edge localization, which is a key property of the nontrivial topological state.

To further verify that the edge state is indeed topological, the Chern number should be calculated. However, since our proposed hyperbolic Haldane model is nonperiodic, the Chern number is undefined in the Brillouin zone torus. In this case, as shown in Fig. 1f, we calculate the real-space Chern number[35,36]

$$C = 12\pi i \sum_{j \in I} \sum_{k \in II} \sum_{l \in III} (P_{jk}P_{kl}P_{lj} - P_{jl}P_{lk}P_{kj}) \quad (2)$$

at each eigenenergy. Here, j, k, and l are site indices in three anti-clockwise regions I, II and III, as shown in the inset of Fig. 1f. The square of projection operator element $|P_{ij}|^2$ measures the correlation of the state density at two sites (i and j) with all eigenstates below the target energy being fully occupied. It is clearly shown that the real-space Chern number around the zero energy possesses a nontrivial value. While, due the finite size effect, the absolute value of calculated real-space Chern number is smaller than 1. The detailed method for the calculation of the real-space Chern number and numerical results for different

lattice sizes are provided in Supplementary Note 2. We note that the nontrivial real-space Chern number clearly manifests the topological property of edge-localized eigenmodes around the zero energy.

In addition, it is widely known that one-way edge states, which are robust against defects, should exist in the energy region with nontrivial real-space Chern numbers. Thus, by solving coupled model equations (see Supplementary Note 3), we numerically study the robust evolution of edge states by launching a wave packet $\psi_{in}(t) = \exp(-(t - t_0)^2/64)\sin(\varepsilon_c t)$ into an edge site as illustrated by the pink arrow in Fig. 1g. Several defects (marked by black triangles) exist at the outermost ring, where the onsite potential is $P_d = 5$ on the defect and it equals to zero on other sites. Other parameters are set as $t_0 = 20$ and $\varepsilon_c = 0.1$. Figure 1g shows the spatial distributions of $|\psi_i(t)|$ at different times with $t = 18, 78, 131$ and $184$, respectively. It is clearly shown that the wave packet unidirectionally moves along the edge of hyperbolic Chern insulators, and passes through defects without backscattering. Additionally, during the propagation, the wave packet is confined to the boundary and does not penetrate into the bulk. These numerical results further prove the existence of unidirectional edge states in the hyperbolic Haldane model. In Supplementary Note 4, numerical results of hyperbolic Chern insulators with different lattice sizes are provided. And, the results of trivial hyperbolic lattice models are also provided in Supplementary Note 5 to further illustrate the difference between topological edge states and trivial edge states. It is worthwhile to note that the ratio of the one-way topological channel (boundary sites) to bulk sites in the {6, 4} hyperbolic Chern insulator is about 0.9 (even with L being infinite), which is much larger than the Euclidean counterpart (approaching to zero in the thermodynamic limit). Hence, such an enhanced topological edge response may improve the efficiency of some topological devices.

Motivated by recent experimental breakthroughs in realizing various quantum phases by circuit networks[36–43], in the following, we design hyperbolic Chern circuits to observe the above proposed novel topological states. Figure 2a illustrates the photograph image of the fabricated circuit sample with $L = 3$. The front and back sides of enlarged views (enclosed the pink dashed block) and the schematic diagram of NN and NNN couplings are plotted in right insets. Specifically, three circuit nodes connected by capacitors $C$ (enclosed by the green block) are considered to form an effective lattice site of the hyperbolic Chern insulator. Voltages at these three nodes are defined by $V_{i,1}$, $V_{i,2}$ and $V_{i,3}$, which could be suitably formulated to construct a pair of pseudospins ($V_{\uparrow i, \downarrow i} = V_{i,1} + V_{i,2}e^{\pm i2\pi/3} + V_{i,3}e^{\mp i2\pi/3}$) for realizing required site couplings. To simulate the real-valued NN hopping rate, three capacitors ($C_\gamma$) framed by the red frame are used to directly link adjacent nodes without a cross. For the realization of NNN hopping rate with a direction-dependent phase ($e^{\pm i2\pi/3}$), three pairs of adjacent nodes are connected crossly via three capacitors $C_\lambda$ (enclosed by the blue block). Each node is grounded by an inductor $L_g$ framed by the white block in the back side. The defect in the outermost ring is achieved by adding an extra grounding capacitor $C_p$. Additionally, boundary nodes should be additionally grounded by suitable capacitors to ensure the same resonance frequency as bulk nodes.

Through the appropriate setting of grounding and connecting, the circuit eigenequation is identical with that of the hyperbolic Chern insulator. Details for the derivation of circuit eigenequations are provided in Supplementary Note 6. In particular, the probability amplitude at the lattice site $i$ is mapped to the voltage of pseudospin $V_{\downarrow, i}$. Amplitudes of the effective NN and NNN couplings equal to $\gamma = C_\gamma/C$ and $\lambda = C_\lambda/C$. The eigenenergy of the hyperbolic Haldane model is directly related to the eigenfrequency of the circuit network as $\varepsilon = f_0^2/f^2 - 3 - 4C_\gamma/C - 8C_\lambda/C$ with $f_0 = (2\pi\sqrt{CL_g})^{-1}$. It is noted that the tolerance of circuit elements is only 1% to avoid the detuning of circuit responses, and circuit parameters are set as $C = 1$ nF, $C_\gamma = 1$ nF, $C_\lambda = 0.2$ nF, $L_g = 1$ uH and $C_P = 5$ nF. Details of the sample fabrication are provided in Methods.

To analyze topological properties of the hyperbolic Chern circuit, we firstly measure the impedance responses of a bulk node (the black line) and an edge node (the red line) using the Wayne Kerr precision impedance analyzer, as plotted in Fig. 2b. We note that the impedance responses are related to the local density of states of the corresponding quantum tight-binding model[38]. It is shown that there are significant impedance peaks of the edge node but neglectable impedances of the bulk node in the frequency range from 1.67 MHz to 1.75 MHz (the red region), which matches to the eigenenergy possessing nontrivial edge states. Figure 2c presents simulation results of frequency-dependent impedance responses using the LTSPICE software. A good consistence between simulations and experiments is obtained, and the larger width of measured impedance peaks results from the lossy effect in the fabricated circuit (see Supplementary Note 7 for details). In addition, the spatial impedance distribution of the circuit at 1.708 MHz is further measured, as shown in Fig. 2d. We can see that the edge-concentrated impedance profile (similar to Fig. 1e with $n = 230$) clearly proves the excitation of hyperbolic edge states.

Then, we measure the temporal dynamics of an effective voltage packet propagating in the hyperbolic Chern circuit. To ensure the excitation of voltage pseudospin, three voltage packets, which are expressed as $[V_{i,1}, V_{i,2}, V_{i,3}] = V(t)[1, \exp(i\frac{2\pi}{3}), \exp(-i\frac{2\pi}{3})]$ with $V(t) = V_0\exp(-(t - t_0)^2/t_d^2)\sin(2\pi f_c t)$, are used. Details of the experimental technologies are provided in Methods. Here, the time delay, packet width and central frequency of the voltage packet are set as $t_0 = 70$ μs, $t_d = 28$ μs and $f_c = 1.708$ MHz, respectively. The voltage packet and the associated frequency spectrum are shown in Fig. 2e, f. The gaussian bandwidth of the input signal is 0.02 MHz. The main components of the frequency spectrum are located in the range sustaining topological edge states, making only nontrivial edge states be excited. Figure 2g, h present time tracks of the voltage packet in the defect-free circuit at two nodes, which are counterclockwise (the red dot) and clockwise (the cyan dot) to the excitation point (the pink arrow) with equal distances. It is clearly shown that only the counterclockwise circuit node possesses a significant response in the time-domain, indicating that the voltage packet propagates counterclockwise along the edge of hyperbolic Chern circuit. Then, we measure the voltage signal at these two nodes with the existence of a defect (the black triangle between the excitation node and the counterclockwise node), as shown in Fig. 2i, j. We can see that the magnitude of voltage packet at the counterclockwise node is nearly unchanged, indicating no significant backscattering appears with the voltage signal passing through the defect. Such a defect-immune voltage propagation clearly manifests the robustness of edge states. The above measurements are also consistent with simulations (see Supplementary Note 8), manifesting the observation of boundary-dominated topological states in hyperbolic Chern circuits.

## Fractal-like higher-order zero modes in deformed hyperbolic lattices

In addition to the boundary-dominated first-order topological states induced by the interplay between the Chern-class topology and the hyperbolic geometry, in the following,

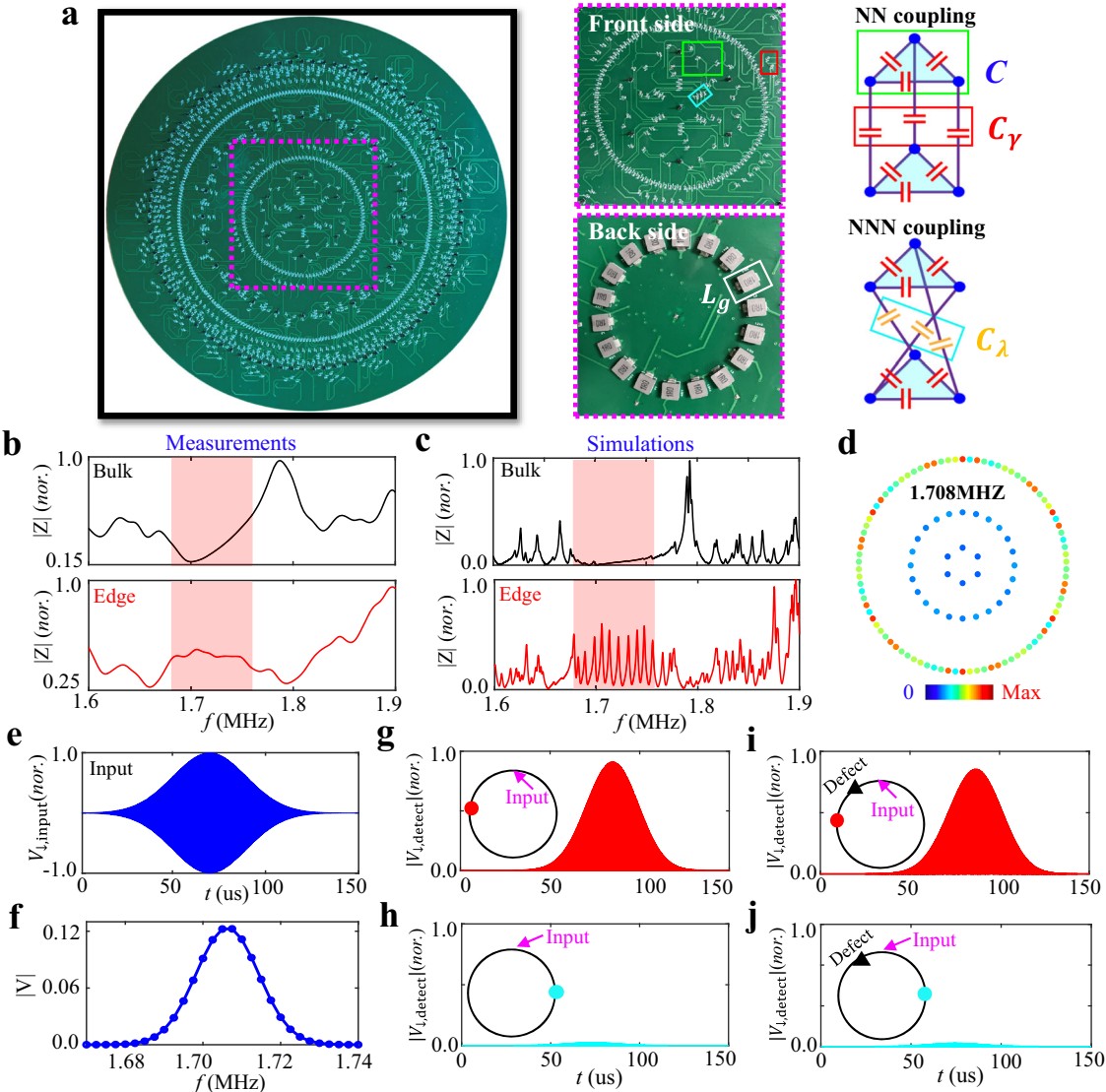

**Fig. 2 Observation of boundary-dominated topological states in the hyperbolic Chern circuit. a** The photograph image of the fabricated hyperbolic circuit. The part of enlarged views and the equivalent schematic diagram are presented in right insets. Three circuit nodes connected by the capacitor $C$ are considered to form an effective lattice site, as enclosed by the green block. The real-valued NN hopping rate is achieved by directly linking adjacent nodes without a cross with three capacitors $C_\gamma$ (framed by the red frame). For the realization of NNN hopping rate with a direction-dependent phase, three pairs of adjacent nodes are connected crossly via three capacitors $C_\lambda$ enclosed by the blue block. Each node is grounded by an inductor $L_g$ framed by the white block in the back side. **b, c** Measured and simulated impedance responses of bulk (black lines) and edge (red lines) nodes. The red region corresponds to the energy range with nontrivial edge states. **d** The measured impedance distributions at 1.708 MHz. **e, f** The voltage packet and the associated frequency spectrum of the injected voltage packet. **g, h** The measured time tracks of the voltage pseudospin at nodes in the defect-free circuit along the counterclockwise and clockwise directions with respect to the excitation node. **i, j** The measured time tracks of the voltage pseudospin at nodes in the circuit with a defect along the counterclockwise and clockwise directions with respect to the excitation node. Red and blue dots in insets mark the position of two detection nodes and the pink arrow presents the input node. The black triangle corresponds to the defect. The circuit parameters used in experiments are set as $C = 1$ nF, $C_\gamma = 1$ nF, $C_\lambda = 0.2$ nF, $L_g = 1$ uH and $C_P = 5$ nF.

we prove that exotic higher-order zero modes can also be constructed in the deformed hyperbolic lattice.

Different from hyperbolic Chen lattices, which possess the complex NNN couplings to break the time-reversal symmetry, here we introduce a pair of distinct coupling strengths ($\gamma_1$ and $\gamma_2$) in the deformed hyperbolic lattice to realize the higher-order zero modes. The general protocol of hyperbolic lattices for achieving higher-order zero modes is presented in Fig. 3a. Six insets present the detailed coupling patterns in different hexagons, which are composed of lattice sites from different layers. In particular, the intralayer coupling strength of the outermost and 2nd (1st and 3rd) layers equals to $\gamma_1$ ($\gamma_2$), as marked by black (pink) lines.

Moreover, the interlayer coupling strength between the nth and (n−1)th layers (n = 4, 3, 2) is identical to the intralayer coupling strength of the nth layer.

At first, we numerically calculate the eigen-spectrum of the deformed hyperbolic lattice with coupling strengths being $\gamma_1 = 1$ and $\gamma_2 = 10$, as shown in Fig. 3b. The inset presents the enlarged eigen-spectrum ranging from −1.5 to 0. To quantify the localization degree of the associated eigenstate, the normalized participation ratio (PR) of each eigenmode $PR = \sum_i |\phi_i(\varepsilon)|^{-4}/N$ is calculated ($N$ is the total number of eigenmodes), as presented by the color bar in Fig. 3b. It is noted that six zero-energy modes

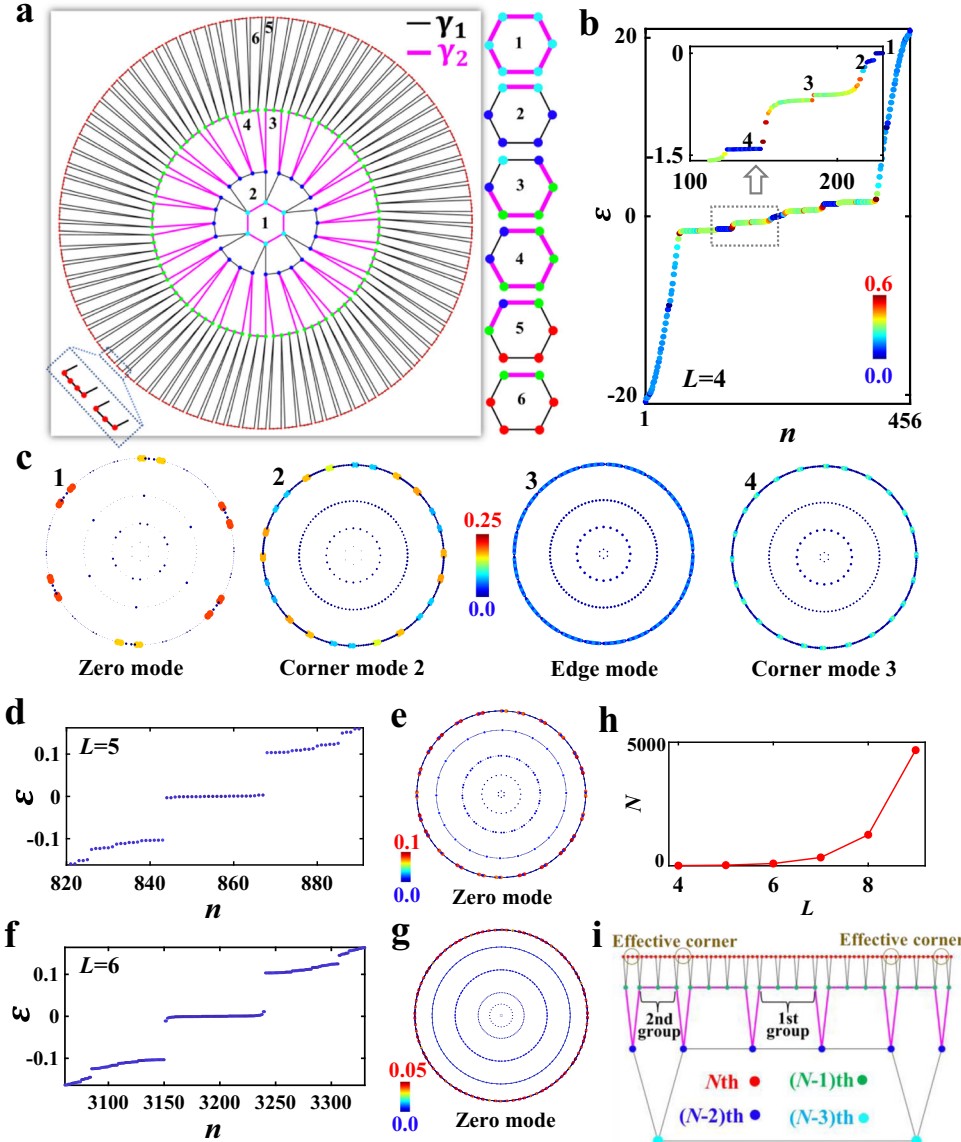

**Fig. 3 Fractal-like higher-order zero modes in the deformed hyperbolic lattice. a** The deformed hyperbolic lattice model for realizing fractal-like higher-order zero modes with $L = 4$ layers. Right insets plot coupling patterns in hexagons composed of lattice sites from different layers. **b** The calculated eigen-spectrum of the system with $L = 4$. The colormap corresponds to the PR of each eigenmode. The inset presents the enlarged spectrum with eigenenergies ranging from $-1.5$ to $0$. **c** Profiles of the normalized density of states of eigenmodes in four different energy regions. **d**, **f** The eigen-spectra of the hyperbolic lattice with $L = 5$ and $L = 6$, respectively. **e**, **g** Density of states of zero modes with $L = 5$ and $L = 6$, respectively. **h** The relationship between the number of midgap zero modes and the layer number ($L$) of deformed hyperbolic lattices. **i** Schematic diagram illustrating the effective corners in the outermost ring in the deformed hyperbolic lattice.

(marked by the number 1 in the inset) show the significant field-localization, which is manifested by the near-zero PR. The PRs of two other kinds of midgap eigenmodes around $\varepsilon = -0.1192$ and $\varepsilon = -1.41$ (marked by numbers 2 and 4) also approach to zero, indicating the strong spatial localizations. In contrast, eigenmodes marked by the number 3 exhibit spatially extended features with larger values of PR. To further illustrate the distribution of each eigenstate, spatial profiles of density of state for these four types of eigenmodes are calculated, as plotted in Fig. 3c. It is clearly shown that the mode distribution of zero-energy eigenstates (labeled by the number 1 and called as the zero mode) is strongly localized around six 0D effective corners, which are different from the real geometric corners in the form of intersections between 1D boundaries. In addition, spatial profiles of eigenmodes with energies being $\varepsilon \sim [-0.1348, -0.1021]$ (labeled by the number 2

and called as the corner mode 2) and $\varepsilon \sim [-1.415, -1.407]$ (labeled by the number 4 and called as the corner mode 3) also present the mode localization at twenty-four effective corners in the outermost ring. In contrast, the averaged spatial profile with $\varepsilon \sim [-0.6154, -0.2025]$ (labeled by the number 3) exhibits the 1D edge localization, where the eigen-fields at corners are nearly zero. These numerical results clearly indicate that the 0D corner-like eigenstates always appear in the gapped edge states.

It is worthwhile to note that these midgap higher-order zero modes in hyperbolic lattices possess similar characteristics to the filling anomaly induced 0D corner states in higher-order topological crystalline insulators[44,45]. In particular, the identical symmetries, including the $C_6$ rotation, the time reversal, and chiral symmetries, are preserved in the deformed hyperbolic lattice as in $C_6$-symmetric higher-order topological crystalline

insulators. In this case, we infer that the nontrivial higher-order zero modes in the gapped edge states should result from obstructed atomic limits in deformed hyperbolic lattices.

While, the required translational symmetry for defining the topological index related to the higher-order topological insulator in Euclidean space[44] is absent in the hyperbolic lattice, making the definition of such a topological invariant become difficult. In Supplementary Note 9, we further illustrate the topological properties of our proposed hyperbolic zero-energy corner states from three aspects. Firstly, we find that the topological phase transition manifested by the closing and reopening of the zero-energy bandgap could be induced by the unbalanced site coupling in the deformed hyperbolic lattice. And, the midgap corner states also appear associated with such a topological phase transition. Moreover, we also show that the topological phase transition appearing in the deformed hyperbolic lattice is similar to the Euclidean counterpart of the $C_6$-symmetric higher-order topological insulator[44,45]. Finally, the robustness of the midgap zero modes in the deformed hyperbolic lattice is also proved, which cannot exist in the interference-induced trivial edge and corner states in hyperbolic lattices sustaining flat bands[22,32]. These features further demonstrate the topological properties of hyperbolic zero modes.

To illustrate the influence of size-dependent boundary geometries of hyperbolic lattices on the formation of higher-order zero modes, we further consider two larger systems. Here, the intralayer coupling strength of the outermost ring ($\gamma_1 = 1$) is always smaller than that of the secondary outer ring ($\gamma_2 = 10$). In Fig. 3d, f, the eigen-spectra of deformed hyperbolic lattices with $L = 5$ and $L = 6$ are calculated. Interestingly, it is clearly shown that there are 24 and 90 midgap zero modes in hyperbolic lattices with L=5 and L=6. Furthermore, the associated spatial profiles of these zero modes are displayed in Fig. 3e, g. Similar to the case with $L = 4$, we note that these zero modes are strongly localized around 24 (for $L = 5$) and 90 (for $L = 6$) corners in the outermost ring, exhibiting the same feature of 0D corner states in higher-order topological insulators. By further calculating the eigenspectra of deformed hyperbolic lattices with different numbers of layers, we plot the relationship between the number of midgap zero modes ($N$) and the total layer number ($L$) in Fig. 3h. It is shown that the number of zero modes increases exponentially with the layer number $L$. The size-dependent mode number is also satisfied for the corner mode 2 and the corner mode 3. It is worthwhile to note that such a phenomenon is similar to the higher-order topological corner states existing in quantum fractals[46], where the number of zero-energy modes depends on the generation number of fractal lattices.

To clarify the fractal-like higher-order zero modes in deformed hyperbolic lattices, the formation mechanism of effective corners in the outermost (the $N$th) ring should be illustrated. As shown in Fig. 3i, lattice sites in the ($N-1$)th ring (green dots) can be divided into two groups. The first (second) group is in the form of the linked chain with four (three) sites. It is noted that the lattice sites in the $N$th layer, which form hexagons combined with lattice sites of the second group in the ($N-1$)th layer, could act as effective corners of zero modes, as enclosed by circles in earth-yellow. In addition, we note that the number of the second group in the ($N-1$)th layer equals to the total number of lattice sites in the ($N-3$)th layer, which increases exponentially with the layer number $L$. In this case, the number of effective corners in the outermost layer of zero modes can also exhibit an exponentially growing tendency with $L$, leading to the appearance of fractal-like higher-order zero modes in the deformed hyperbolic lattice.

Similar to the hyperbolic Chern insulator, we can also design circuit networks to observe hyperbolic higher-order zero modes.

Figure 4a illustrates the photograph image of the fabricated circuit with $L = 4$, and the enlarged views of front and back sides of the sample (marked by the red dash block) are plotted in right insets. In particular, the coupling strength of $\gamma_1$ ($\gamma_2$) in the hyperbolic lattice model is realized by linking circuit nodes through the capacitor $C_1$ ($C_2$), as framed by the blue (green) circle. And, each circuit node is grounded by an inductor $L_{gc}$ enclosed by the white block. Moreover, boundary sites should be additionally grounded by two capacitors $C_2$ to ensure the same resonance frequency as bulk nodes. In this case, the circuit eigenequation is identical with that of the deformed hyperbolic lattice (see Supplementary Note 10 for details), and the eigenenergy of the hyperbolic lattice is directly related to the eigenfrequency of the circuit as $\varepsilon = f_c^2/f^2 - 2 - 2C_1/C_2$ with $f_c = 1/2\pi\sqrt{C_2 L_{gc}}$. Here, circuit parameters are set as $C_1 = 1$ nF, $C_2 = 10$ nF and $L_{gc} = 3.3$ uH, and the tolerance of those circuit elements is limited by 1%.

To observe higher-order corner modes, we measure site-resolved impedance responses of selected bulk, edge and corner nodes, as shown in Fig. 4b. The corresponding numerical results are presented in Fig. 4c, where measured results are consistent with simulations and wider peaks in experiments result from lossy effects in the sample. It is clearly shown that the corner node possesses significant impedance peaks in three frequency ranges marked by red regions. In particular, the frequency of the impedance peak in the central red region (around 0.591 MHz) matches to the zero-energy, indicating the excitation of higher-order zero modes. It is worthy to note that due to the extremely small spectral distance between zero modes and 'corner mode 2', the impedance peaks induced by these two kinds of corner states are merged together. Additionally, other two peaks of the corner node are originated from the resonance of 'corner mode 3' with positive and negative energies. Additionally, measured impedances of bulk and edge nodes are relatively small in frequency regions sustaining corner impedance peaks. This phenomenon clearly shows that the corner modes always exist in bandgaps of bulk and edge states. The edge node shows large impedance peaks in four frequency regions marked by blue regions, which are consistent with the calculated eigen-energies of edge states in Fig. 3b (and the chiral symmetric counterpart with positive energies). The bulk node possesses many little impedance peaks in low-frequency and high-frequency regions, as shown in black regions, corresponding to the resonance peaks associated to bulk modes.

To obtain the spatial distribution of hyperbolic higher-order corner modes, we further recover the circuit admittance spectrum (see Methods for details), and illustrate the spatial distributions of three different higher-order 0D corner modes of the recovered circuit Laplacian, as shown in Fig. 4d–f. We can see that the recovered mode distributions are consistent with spatial profiles of higher-order corner modes in Fig. 3c. These experimental results clearly prove that the hyperbolic higher-order corner modes have been fulfilled in our designed circuit network.

**Discussion**. We report the experimental observation of boundary-dominated first-order Chern edge states and fractal-like higher-order zero modes in hyperbolic circuit networks. By extending the definition of the Haldane model to hyperbolic spaces, a unidirectional edge state with the nontrivial real-space Chern number is proposed. Besides the first-order topological state, the fractal-like midgap higher-order zero modes are also revealed based on the deformed hyperbolic lattice with unequal coupling strengths in different layers. The physical origin for the appearance of these exotic topological states results from the

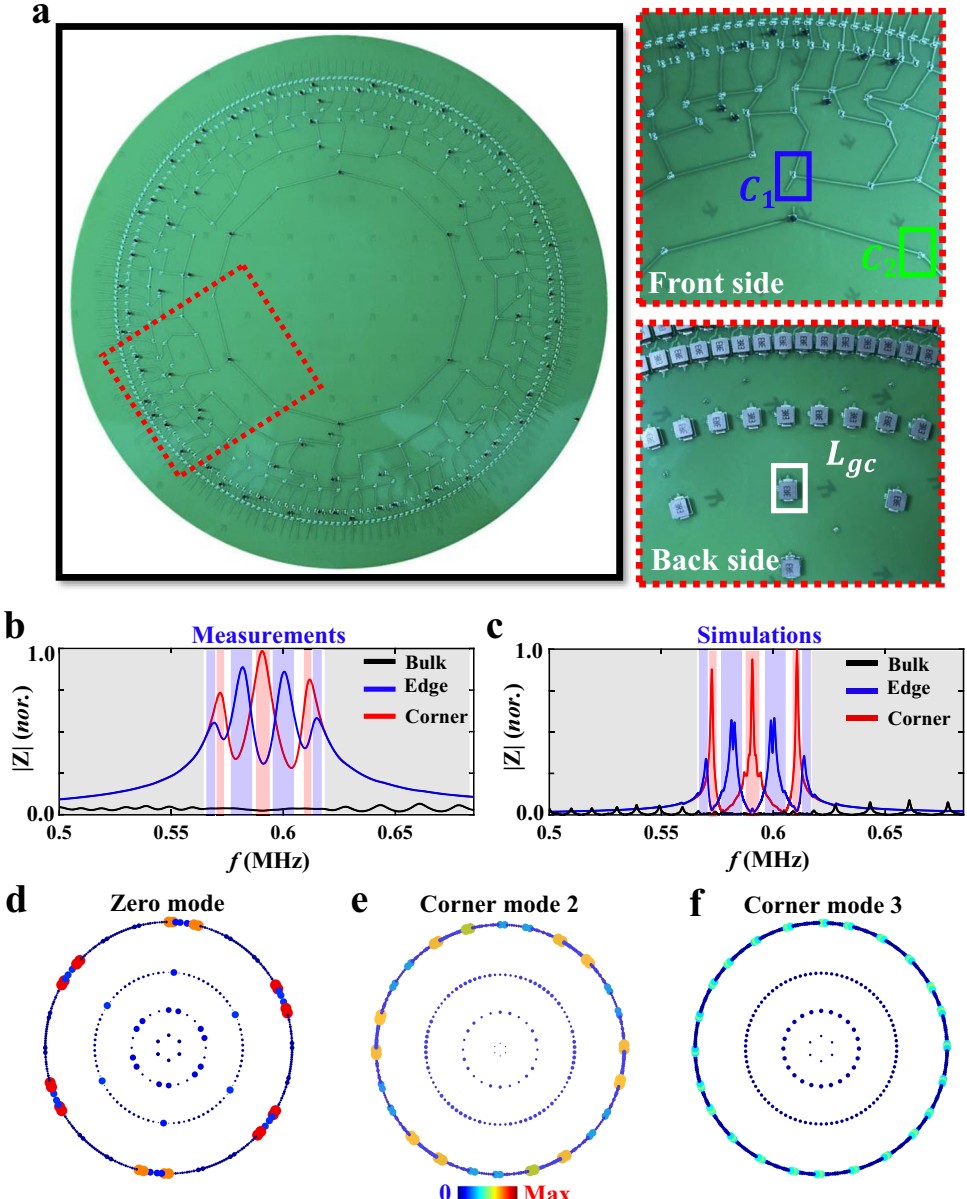

**Fig. 4 Observation of higher-order zero modes in the hyperbolic circuit. a** Photograph image of the fabricated hyperbolic circuit. Right insets display the enlarged view of the sample as well as the associated schematic diagram. The coupling strength of $\gamma_1$ ($\gamma_2$) in the hyperbolic lattice model is realized by linking circuit nodes through the capacitor $C_1$ ($C_2$), as framed by the blue (green) circle. And, each circuit node is grounded by an inductor $L_{gc}$ enclosed by the white block. **b**, **c** Measured and simulated impedance responses of selected bulk (black lines), edge (blue lines) and corner (red lines) nodes. **d**–**f** The profiles of higher-order zero modes and other two 0D corner states of the recovered circuit Laplacian. Circuit parameters used in experiments are set as $C_1 = 1$ nF, $C_2 = 10$ nF, and $L_{gc} = 3.3$ uH.

negative curvature induced boundary effect, where boundary sites always occupy a finite portion of the total site regardless of the system size. These fascinating topological states are observed in experiments by hyperbolic circuit networks. Compared with topological states in Euclidean space, one significant feature of the hyperbolic counterpart is that the edge lattice sites acting as the topological channel always occupy a large ratio of total lattice sites even at the thermodynamic limit. Hence, by incorporating the proposed hyperbolic topological states into the design of robust functional devices in other classical wave systems, such as topological lasers, the operational efficiency and spatial utilization may be remarkably improved. Our proposal provides a flexible platform to further investigate and visualize

more interesting phenomena related to topological physics in hyperbolic lattices.

With the flexibility that the connection and grounding of circuit nodes are allowed in any desired way free from constraints of locality and dimensionality, the high-dimensional topological hyperbolic lattice with non-local site couplings could also be achieved. Moreover, including nonreciprocal, non-Hermitian and non-linear elements into the hyperbolic circuit network, the novel behavior induced by the interplay between the non-Hermitian, the non-linearly, topologies and the curvature can be investigated in experiments. Finally, the designed circuit simulator could also give a new way to manipulate the electronic signal with exotic behaviors.

## Methods

**Sample fabrications and circuit measurements**. We exploit electric circuits by using LCEDA program software, where the PCB composition, stack-up layout, internal layer and grounding design are suitably engineered. Here, the PCBs possessing the Chern edge states and higher-order zero modes have six and four layers, respectively, where two layers are used for the inner electric layer and the node couplings are arranged in the remaining layers. It is worth noting that the all grounded components are grounded through blind buried holes. Moreover, all PCB traces have a relatively large width (0.75 mm) to reduce the parasitic inductance, and the spacing between electronic devices is also large enough to avert spurious inductive coupling. The SMP connectors are welded on the PCB nodes for the signal input. To ensure the tolerance of circuit elements and series resistance of inductors to be as low as possible, we use a WK6500B impedance analyzer to select circuit elements with high accuracy (the disorder strength is only 1%) and low losses.

For the time-domain measurement, we use two signal generators (DG5072) to inject three designed wave packets with required initial phases for exciting the voltage pseudospin. One output of the signal generator (the initial phase is set to 0) is directly connected to one end of the oscilloscope (Agilent Technologies Infiniivision DSO7104B) to ensure an accurate start time. The scanning speed of oscilloscope is set as 10 ms/s. The measured voltage signals are in the range from 0 μs to 200 μs in the time domain, where 0 μs is defined as the time for the simultaneous signal injection and measurement.

Recovering the circuit admittance spectrum involves a series of operations, where a current is injected at each circuit node individually and the voltages at all circuit nodes are measured at the same time. Based on the measured voltages and input currents, we can obtain Green's function of the hyperbolic circuit, which is the inverse of the circuit Laplacian. Calculating the eigenvalues and eigenvectors of the recovered circuit Laplacian, the admittance eigen-spectrum and the associated mode profiles are obtained.

## Data availability

All data are displayed in the main text and Supplementary Information. The data that support the findings of this study are available from the corresponding author upon reasonable request.

## Code availability

The code that supports the plots within this paper is available from the corresponding author upon reasonable request.

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

## Acknowledgements

W.Z., H.Y., N.S., H.S., and X.Z. are supported by the National Key R & D Program of China under Grant No. 2017YFA0303800 and the National Natural Science Foundation of China (No. 91850205 and No.12104041).

## Author contributions

W.Z. finished the theoretical scheme and designed the circuit simulator. H.Y. and N.S. finished the experiments with the help of H.S., W.Z., and X.Z. wrote the manuscript. X.Z. initiated and designed this research project.

## Competing interests

The authors declare no competing interests.
