## [Peer Review File · Nature Communications]

REVIEWER COMMENTS

Reviewer #1 (Remarks to the Author):

In this manuscript, Zhang et al. investigated topological states in hyperbolic lattices using circuit platforms. The authors explored topological states in the Haldane model and zero modes in deformed lattices. For both phenomena, the authors conducted in-depth theoretical (real-space Chern number and fractal-like grouping), numerical (tight-binding, coupled mode, and circuit eigenequation), and experimental (circuit network) demonstration. The analysis results of the manuscript were well-developed and supported one another.

The critical impact of this manuscript is on the first experimental demonstration of topological hyperbolic lattices, in comparison with the first experimental work on hyperbolic lattices [ref. 7] and the first theoretical work on topological hyperbolic lattices [ref. 32]. When considering the increasing interest in exploiting non-Euclidean geometry to wave phenomena, the manuscript handles a timely issue. Furthermore, the studies of the manuscript experimentally verified intriguing features of topological states (edge dominance) and zero modes (enhanced degeneracy) in hyperbolic geometry compared with Euclidean ones for the first time, providing the fertile evidences to their claim. Therefore, the manuscript will provide a critical contribution in photonics, acoustics, electronic circuits, and related fields in terms of accessing non-Euclidean degrees of freedom and their interactions with topological phenomena.

However, for the publication with significant impact on the related research fields, the current manuscript should be revised thoroughly due to some critical issues on the flow of the manuscript, the review of previous works, and insufficient analysis and details of their results (see below for details). Therefore, the reviewer suggests the major revision of the manuscript.

1. In the manuscript, Figs. 1 and 2 treat topological states in topological hyperbolic lattices developed from the Haldane model, while Figs. 3 and 4 handle zero modes originating from deformed hyperbolic lattices (weaker coupling in the outer region of the system). While the former and latter discussion is nearly disconnected, the demonstration of the topological nature of zero modes is absent in the manuscript, though some phenomenological description of the similarity with corner states in higher-order topological crystalline insulators was included. In order to coherently describe the authors' work with a viewpoint based on topological states, more efforts on demonstrating topological properties of the observed corner states should be included, for example, differentiating topological corner states from edge states or trivial corner states in hyperbolic lattices.

2. In a similar context, as described in [ref. 7] or Supplementary Materials of [ref. 32], hyperbolic lattices inherently lead to enhanced degeneracies, which may lead to trivial edge or corner states due to the interference between the degenerate eigenmodes of the same energy (or flat band systems). The verification of topological natures of the zero modes in Figs. 3 and 4 is thus essential to support the authors' main claim.

3. While the first theoretical discovery on topological states using the non-Euclidean generalization of the Landau gauge was shown in [ref. 32], the introduction part of the manuscript does not fairly review this previous work. More detailed review and comparison with [ref. 32] and the following work [arXiv:2111.05779] are necessary. For example, the Haldane model allows for more direct (or Euclidean-like) assignment of the gauge field and Berry curvature (in contrast to the tree-like design of the Landau gauge in [ref. 32]) but is difficult to be realized in high-frequency regimes such as photonics due to the next nearest neighbor couplings.

4. The authors stated that the given finite hyperbolic lattice is topologically equivalent with successive quasi-concentric rings. The reviewer has difficulty in understanding this statement because in terms of the concentric rings, the hyperbolic lattice is apparently composed of disconnected lines with inter-connection between rings, which should be

topologically different from simple concentric rings. More comprehensible descriptions about "quasi-"concentric rings are thus required.

5. In understanding the results in Fig. 2, the theoretical comparisons with trivial cases of (i) NN coupling ($\lambda = 0$) and (ii) NN & NNN couplings ($\varphi = 0$ with reciprocal or non-reciprocal coupling) are certainly helpful. Because the geometry of hyperbolic lattices has edge-dominant features, there usually exist trivial edge states, as demonstrated in [ref. 32].

Therefore, the comparison with trivial cases, for example, by using the calculation of the real-space Chern number, will enhance the novelty and impact of the authors' findings.

6. Several notations are confusing. For example, the letter φ was employed to express different physical quantities: the eigenmode $\varphi(\varepsilon)$, temporal field evolution ($\varphi(t)$, which is the superposition of the eigenmode), and gauge field φ . All the quantities should be distinguished.

7. The definition of P_d for the on-site energy and the value of original on-site energy are missing.

8. The input wave packet is not related well to the band of hyperbolic lattices. The authors may depict the gaussian bandwidth of the input signal in Fig. 2d.

9. Although the authors stated that the edge confinement (0.9) of one-way topological states is much larger than that of the Euclidean counterpart, the value for the Euclidean one is absent.

10. The compactness may not be the advantage of hyperbolic lattices because the underlying physics of hyperbolic lattices requires more complex and intense coupling than the Euclidean one. The edge confinement requires such complex geometry despite the suppression of the field inside the bulk region.

11. For general readers in topological wave mechanics, it may be helpful to include the discussion on the meaning of the impedance peaks, which corresponds to the transmission peaks in topological photonics or acoustics.

12. While the authors stated that the broadening in experimental results originates from the loss of elements, the theoretical results, including the loss in each element, may be included in Supplementary Materials for the completeness of the manuscript.

Reviewer #2 (Remarks to the Author):

The authors propose theoretically and realize experimentally an analog of the well-known Haldane model of a Chern insulator on a hyperbolic lattice. By numerical diagonalization in a disk geometry, the authors show the energy spectrum contains a spectral region near zero energy where eigenstates have dominant support on the sample boundary. The authors then calculate an energy-dependent real-space Chern number, and show that this Chern number exhibits a quantized plateau in roughly the same region where boundary states exist. A wave packet injected near the sample edge is shown to propagate unidirectionally and in a manner undisturbed by defects on the boundary.

The theoretical predictions are then tested experimentally on an electric circuit network. The physical system exhibits lossy (resistive) behavior that is not present in the theoretical model, but the agreement between theory and experiment is altogether good notwithstanding. In particular, the measurements demonstrate the key features expected of a Chern insulator: localized edge states within a bulk gap, chiral edge propagation, and protection against backscattering. Finally, the authors also argue that with suitable modifications, the model Hamiltonian and experimental system can support corner modes indicative of higher-order topology.

In my opinion, this is an important work that produces two key advances: a theoretical

generalization of the Haldane model to hyperbolic space, and its experimental simulation using classical circuit networks. The connection between the tight-binding model and the classical circuit equations is well explained, and the experimental demonstration compelling (within the unavoidable limitation of small system sizes). The relevant literature is appropriately cited, and this work will significantly advance the burgeoning field of exotic "hyperbolic matter". (The additional discussion of higher-order topology is somewhat less convincing, in my opinion, but I discuss this further below.)

Before I can recommend publication, I would like the authors to address the following issues/questions.

- Although the evidence from the real-space Chern number is compelling, one would like to further check that the presence of edge states is really due to the topology of the model, as opposed to the unusual boundary/bulk ratio in the Poincaré disk, which is purely a geometrical effect. To that end, could the authors also present theoretical results in the "trivial insulator" phase, e.g., for a value of the NNN phase φ for which the real-space Chern number is zero? Ideally, one would like to see that in such a trivial insulator, there are no edge states near zero energy, and that a wave packet injected near the boundary spreads out into the bulk instead of propagating unidirectionally along the boundary. In my opinion, such a comparison between trivial and topological regimes would significantly strengthen the authors' claims.
- Where Fig. 2b-c are discussed in the text, the discussion seems to imply that the measured impedance spectrum is simply related to the local density of states (LDOS) of the corresponding quantum tight-binding model. Is there such a relation? If so, it would be nice to have a mention and/or derivation of this (e.g. in the supplementary material).
- The part of the paper I find somewhat less clear/compelling is the discussion of higher-order topological modes. First, it is not clear how the authors arrived at the model with two different coupling strengths γ_1 and γ_2 . Is this motivated by a Euclidean equivalent? This should be further discussed.
- Related to the previous point, I am not sure if the corner modes are really a consequence of bulk topology, as opposed to trivial boundary modes arising from a cleverly engineered confining potential. It looks like the dependence of couplings on the layer index n implies the model is not translationally invariant away from the boundaries. If the authors' model were extended to the infinite lattice $L \rightarrow \infty$, would it have the full periodicity of the {6,4} tiling?
- In Fig. 1d and/or the corresponding text, the authors should explain that the color scale corresponds to $V(\epsilon)$, which is not clear in the current version.
- For the real-space Chern number Eq. (2), the authors should rather cite the original reference by A. Kitaev, Ann. Phys. 321, 2 (2006).
- In the discussion after Eq. (6) in the SM, perhaps it would be best to leave out "symmorphic". I believe hyperbolic triangle groups do not generally have a semidirect product decomposition, though this may have to be checked.
- In Fig. S3a, the times indicated on the figure do not match those quoted in the text and the figure caption.
- In Eqs. (9-12) in the SM, it would be preferable to use brackets (...) or [...] for matrices and vectors, instead of |...|, since the latter can be confused as a determinant. Also, Eq. (9) seems to be missing a closing parenthesis ").
- There are several typos in the text: "reach" -> "research" (p. 2); "observe above" -> "observe the above" (p. 6); "impedence" -> "impedance" several times throughout the manuscript; "eigenspectral" -> "eigenspectra" (p. 12); "remained" -> "remaining" (p. 15); "atom limit" -> "atomic limit" (p. 9 of SM). Also, the phrase "with extremely fewer trivial regions" at the end of the abstract is rather unclear. I would suggest something like "which maximize the topological edge response".

Response Letter to Reviewers

We are grateful for the constructive comments on this manuscript (NCOMMS-22-06101-T) from two reviewers.

In the text below, reviewer comments are quoted in **blue** and followed by our detailed response. We have also revised the manuscript and the Supplemental Materials based on the reviewer comments, and these updates are highlighted in **red** in those files. In the text below, these updates are also highlighted in *Italics*.

Response to comments of the reviewer #1

In this manuscript, Zhang et al. investigated topological states in hyperbolic lattices using circuit platforms. The authors explored topological states in the Haldane model and zero modes in deformed lattices. For both phenomena, the authors conducted in-depth theoretical (real-space Chern number and fractal-like grouping), numerical (tight-binding, coupled mode, and circuit eigenequation), and experimental (circuit network) demonstration. The analysis results of the manuscript were well-developed and supported one another.

The critical impact of this manuscript is on the first experimental demonstration of topological hyperbolic lattices, in comparison with the first experimental work on hyperbolic lattices [ref. 7] and the first theoretical work on topological hyperbolic lattices [ref. 32]. When considering the increasing interest in exploiting non-Euclidean geometry to wave phenomena, the manuscript handles a timely issue. Furthermore, the studies of the manuscript experimentally verified intriguing features of topological states (edge dominance) and zero modes (enhanced degeneracy) in hyperbolic geometry compared with Euclidean ones for the first time, providing the fertile evidences to their claim. Therefore, the manuscript will provide a critical contribution in photonics, acoustics, electronic circuits, and related fields in terms of accessing non-Euclidean degrees of freedom and their interactions with topological phenomena.

However, for the publication with significant impact on the related research fields, the current manuscript should be revised thoroughly due to some critical issues on the flow of the manuscript, the review of previous works, and insufficient analysis and details of their results (see below for details). Therefore, the reviewer suggests the major revision of the manuscript.

Reply: We would like to thank the reviewer for the careful review, positive evaluation and valuable suggestions of our work. In the following, we will give a detailed response to all points proposed by the reviewer.

1. In the manuscript, Figs. 1 and 2 treat topological states in topological hyperbolic lattices developed from the Haldane model, while Figs. 3 and 4 handle zero modes originating from deformed hyperbolic lattices (weaker coupling in the outer region of the system). While the former and latter discussion is nearly disconnected, the demonstration of the topological nature of zero modes is absent in the manuscript, though some phenomenological description of the similarity with corner states in higher-order topological crystalline insulators was included. In order to coherently describe the authors' work with a viewpoint based on topological states, more efforts on

demonstrating topological properties of the observed corner states should be included, for example, differentiating topological corner states from edge states or trivial corner states in hyperbolic lattices. 2. In a similar context, as described in [ref. 7] or Supplementary Materials of [ref. 32], hyperbolic lattices inherently lead to enhanced degeneracies, which may lead to trivial edge or corner states due to the interference between the degenerate eigenmodes of the same energy (or flat band systems). The verification of topological natures of the zero modes in Figs. 3 and 4 is thus essential to support the authors' main claim.

Reply: We would like to thank the reviewer for the comment. The above two comments are all about the demonstration of topological properties of hyperbolic zero-modes. In the following, we illustrate the topological properties of our proposed hyperbolic zero-energy corner states from three aspects.

Firstly, we focus on the topological phase transition induced by the unbalanced site coupling in the deformed hyperbolic lattice. As shown in Figs. R1a-R1c, we plot the calculated eigen-spectra of the deformed hyperbolic lattice with the ratio of γ_1/γ_2 being 10, 1 and 0.1, respectively. The color bar corresponds to the participation ratio (PR) of each eigen-mode. We set $L=6$, where the influence of finite size effect could be neglected. It is clearly shown that when the coupling strength in the outermost layer is the larger one ($\gamma_1/\gamma_2=10$), the edge states are gapped around the zero energy, and there is no midgap corner state. In such a case, this bandgap is a trivial gap. The topological phase transition could appear with closing and reopening of the bandgap. It is shown that the gap is closing (around the zero-energy) with balanced values of γ_1 and γ_2 ($\gamma_1/\gamma_2=1$), corresponding to the original $\{6,4\}$ hyperbolic lattice. By further decreasing the ratio to the case with a smaller coupling strength in the outermost layer ($\gamma_1/\gamma_2=0.1$), the gap of edge states is reopened, and the midgap corner modes appear. Such a gap closing and reopening phenomenon associated with the appearance of midgap corner states is a convenient evidence for the topological phase transition.

Fig. R1. The calculated eigen-spectra of the deformed hyperbolic lattice with the ratio between γ_1 and γ_2 being 10, 1 and 0.1, respectively. Here, we set $L=6$. The color bar corresponds to the PR of each eigen-energy.

Next, we show that the topological phase transition appearing in the deformed hyperbolic lattice is similar to the Euclidean counterpart of the C_6 -symmetric higher-order topological insulator (Ref. 44 and Ref. 45). To clearly prove this similarity, we consider the C_6 -symmetric higher-order topological insulator, as shown in Fig. R2a, where the different values of intra-cell (γ_1) and inter-

cell (γ_2) couplings exist. In the following calculations, the finite C_6 -symmetric $\{6,3\}$ lattice containing 37 units is considered. As shown in Fig. R2b-R2d, we calculate the eigen-spectra of the system with the ratio between γ_1 and γ_2 being 10, 1 and 0.1, respectively. We can see that, by tuning the ratio of intra- and inter-cell couplings, the topological phase transition accompanied with the closing and reopening of bandgap happens. And, the midgap corner state could appear in the non-trivial bandgap. This phenomenon is identical to the above discussed hyperbolic topological phase transition. In this case, due to the same topological phase transition and identical symmetries (the C_6 rotation, the time reversal, and chiral symmetries), we deduce that midgap higher-order zero modes in deformed hyperbolic lattices possess similar characteristics to the filling anomaly induced 0D corner states in C_6 -symmetric higher-order topological crystalline insulators.

Fig. R2. (a) The schematic diagram of the C_6 -symmetric higher-order topological insulator, where the intra-cell (γ_1) and inter-cell (γ_2) couplings exist. (b)-(d) The eigen-spectra of the C_6 -symmetric lattice with the ratio between γ_1 and γ_2 being 10, 1 and 0.1, respectively. Here, in the calculation, the finite C_6 -symmetric lattice contains 37 units.

Fig. R3. (a) and (b) The calculated eigen-spectra with disorder strength being $W=0.1$ and $W=0.2$, respectively. Other parameters are set as $\gamma_1=1$, $\gamma_2=10$ and $L=6$. (c) The zero-mode distribution with $W=0.2$.

Finally, to further differentiate topological corner states from edge states or trivial corner states, we investigate the robustness of the midgap zero modes in the deformed hyperbolic lattice. Here, we introduce a little disorder to the onsite potential $[-W, W]$ at all bulk sites and edge sites, and keep the onsite potential of corner sites unchanged. Other parameters are set as $\gamma_1=1, \gamma_2=10$ and $L=6$. Figs. R3a and R3b plot the calculated eigen-spectra with W being 0.1 and 0.2, respectively. The colormap corresponds to the PR. We can see that the random onsite potential could alter the eigen-energies and localizations of trivial edge states. While, the midgap zero modes are always fixed, and the corresponding mode distribution, as shown in Fig. R3c with $W=0.2$, also keeps the same. These numerical results manifest the robustness of hyperbolic zero modes. Because, disorders should break the enhanced energy degeneracies and interference effects in the trivial hyperbolic lattice, which makes the robust property cannot exist in the interference-induced trivial edge and corner states in hyperbolic lattices sustaining flat bands.

Action taken:

In the revised manuscript, we have added the following discussion in page 12 to illustrate the topological properties of hyperbolic zero modes: *“While, the required translational symmetry for defining the topological index related to the higher-order topological insulator in Euclidean space⁴⁴ is absent for the hyperbolic lattice. This makes the definition of a topological invariant for the higher-order zero mode become difficult. In Supplementary Note 9, we further illustrate topological properties of our proposed hyperbolic zero-energy corner states from three aspects. Firstly, we find that the topological phase transition manifested by the closing and reopening of the zero-energy bandgap associated with the appearance of midgap corner states could appear by tuning the unbalanced site coupling in the deformed hyperbolic lattice. Moreover, we also show that the topological phase transition appearing in the deformed hyperbolic lattice is similar to the Euclidean counterpart of the $C6$ -symmetric higher-order topological insulator^{44,45}. Finally, the robustness of the midgap zero modes in the deformed hyperbolic lattice is also proved, which cannot exist in the interference-induced trivial edge and corner states in hyperbolic lattices sustaining flat bands^{22, 32}. These features further demonstrate the topological properties of hyperbolic zero modes.”*

- In the Supplementary Note 9, we have added numerical results to demonstrate the topological properties of higher-order zero modes of deformed hyperbolic lattices.

3. While the first theoretical discovery on topological states using the non-Euclidean generalization of the Landau gauge was shown in [ref. 32], the introduction part of the manuscript does not fairly review this previous work. More detailed review and comparison with [ref. 32] and the following work [arXiv:2111.05779] are necessary. For example, the Haldane model allows for more direct (or Euclidean-like) assignment of the gauge field and Berry curvature (in contrast to the tree-like design of the Landau gauge in [ref. 32]) but is difficult to be realized in high-frequency regimes such as photonics due to the next nearest neighbor couplings.

Reply: We would like to thank the reviewer for the kind suggestion. In the revised manuscript, we have added the following discussion in the introduction part to review these important works:

- *“Recently, the hyperbolic topological state has been theoretically proposed based on a tree-like design of the Landau gauge in periodic and open systems^{32,28}.”*
- *“We note that the Haldane model allows for more direct (or Euclidean-like) assignment of the gauge field and Berry curvature compared to the tree-like design of the Landau gauge³², but it*

is difficult to be realized in high-frequency regimes (such as photonics) due to the requirement of next nearest neighbor couplings. Hence, in experiments, the suitably designed circuit network, where the long-range site coupling is easily to be realized, is used to construct the hyperbolic Haldane model.”.

4. The authors stated that the given finite hyperbolic lattice is topologically equivalent with successive quasi-concentric rings. The reviewer has difficulty in understanding this statement because in terms of the concentric rings, the hyperbolic lattice is apparently composed of disconnected lines with inter-connection between rings, which should be topologically different from simple concentric rings. More comprehensible descriptions about “quasi-“concentric rings are thus required.

Reply: We would like to thank the reviewer for the comment. The property of hyperbolic tight-binding lattice model depends on the connection pattern of all vertices, and is regardless of the configuration of all vertices. In this case, the hyperbolic lattice model could also be illustrated by arranging the vertices in the hyperbolic lattice to quasi-concentric rings, and maintaining the connection pattern of all vertices unchanged.

Action taken:

- In the revised manuscript, we have added the following discussion in page 4 to give a more comprehensible descriptions about quasi-“concentric rings: *“It is noted that the property of hyperbolic tight-binding lattice model depends on the connection pattern of all vertices, and is regardless of the configuration of all vertices. Hence, the hyperbolic lattice model could also be illustrated by arranging the vertices in the form of quasi-concentric rings, and maintaining the connection pattern of all vertices unchanged. In this case, the finite hyperbolic lattice {6, 4} with a sixfold rotation invariance in Poincaré disk (shown in Fig. 1b) is equivalent to successive quasi-concentric rings, as shown in Fig. 1c, with $L=4$ layers.”.*

5. In understanding the results in Fig. 2, the theoretical comparisons with trivial cases of (i) NN coupling ($\lambda = 0$) and (ii) NN & NNN couplings ($\varphi = 0$ with reciprocal or non-reciprocal coupling) are certainly helpful. Because the geometry of hyperbolic lattices has edge-dominant features, there usually exist trivial edge states, as demonstrated in [ref. 32]. Therefore, the comparison with trivial cases, for example, by using the calculation of the real-space Chern number, will enhance the novelty and impact of the authors’ findings.

Reply: We would like to thank the reviewer for the comment. We calculate the eigen-spectra, the corresponding real space Chern numbers, and the dynamics of injected wave packet for systems without NNN couplings $\lambda = 0$ (shown in Fig. R4), and with real-valued NNN couplings $\lambda = 0.2$ and $\varphi = 0$ (shown in Fig. R5). We can see that the real-space Chern number is trivial around the zero-energy, and the one-way propagation of edge state is also absence.

Fig. R4. (a)-(c) The calculated eigen-spectrum, the real space Chern numbers, and the dynamics of injected wave packet for systems with the NNN coupling being $\lambda = 0$. The color bar corresponds to the localization degree at the boundary.

Fig. R5. (a)-(c) The calculated eigen-spectrum, the real space Chern numbers, and the dynamics of injected wave packet for systems with parameters being $\lambda = 0.2$ and $\varphi = 0$. The color bar corresponds to the localization degree at the boundary.

Action taken:

- In the revised manuscript, we have added the following discussion in page 5: “*And, the results of trivial hyperbolic lattice models are also provided in Supplementary Note 5 to further illustrate the difference between topological edge states and trivial edge states.*”.
- In the Supplementary note 5, we have added numerical results of the trivial hyperbolic lattice model to further illustrate the difference between topological edge states and trivial edge states.

6. Several notations are confusing. For example, the letter ϕ was employed to express different physical quantities: the eigenmode $\phi(\epsilon)$, temporal field evolution ($\phi(t)$, which is the superposition of the eigenmode), and gauge field ϕ . All the quantities should be distinguished.

Reply: We would like to thank the reviewer for the comment. We re-define these quantities, where $\phi_i(\epsilon)$ corresponds to the eigen-mode, $|\psi_i(t)|$ corresponds to the temporal field evolution, and ϕ represents the gauge field.

7. The definition of P_d for the on-site energy and the value of original on-site energy are missing.

Reply: We would like to thank the reviewer for the comment. In page 5 of the revised manuscript, we have added the following discussion to illustrate the distribution of the on-site energy: “*The onsite potential is $P_d=5$ on the defect, and it equals to zero on other sites.*”.

8. The input wave packet is not related well to the band of hyperbolic lattices. The authors may depict the gaussian bandwidth of the input signal in Fig. 2e.

Reply: We would like to thank the reviewer for the comment. The frequency spectrum of the injected voltage packet is shown in Fig. 2f. The gaussian bandwidth and the central frequency of the input signal (in Fig. 2e) are 0.02MHz and 1.708MHz, respectively. And the frequency-range from 1.67 MHz to 1.75 MHz corresponds to the eigenenergy possessing nontrivial edge states (determined by the formula $\epsilon = f_0^2/f^2 - 3 - 4C_\gamma/C - 8C_\lambda/C$). Hence, the frequency spectrum of input voltage packet is located in the range sustaining topological edge states, making only nontrivial edge states be excited.

Action taken:

- In page 9 of the revised manuscript, we have added the following discussion to depict the gaussian bandwidth of the input signal: “*The gaussian bandwidth of the input signal is 0.02MHz. The main components of the frequency spectrum are located in the range sustaining topological edge states, making only nontrivial edge states be excited.*”.

9. Although the authors stated that the edge confinement (0.9) of one-way topological states is much larger than that of the Euclidean counterpart, the value for the Euclidean one is absent.

Reply: We would like to thank the reviewer for the comment. Boundary sites always occupy a finite portion of the total site regardless of the size for the hyperbolic lattice. This is completely contrary to the case of Euclidean lattices, where the ratio between the number of boundary sites to the total sites approaches to zero in the thermodynamic limit. This effect makes the edge confinement of one-way hyperbolic edge states could be much larger than that of the Euclidean counterpart (*approaching to zero in the thermodynamic limit*).

Action taken:

- In the page 5 of the revised manuscript, we have added the following discussion to illustrate the difference of edge confinements between hyperbolic and Euclidean counterparts: “*It is*

worthwhile to note that the ratio of the one-way topological channel (boundary sites) to bulk sites in the {6, 4} hyperbolic Chern insulator is about 0.9 (even with L being infinite), which is much larger than the Euclidean counterpart (approaching to zero in the thermodynamic limit).”.

10. The compactness may not be the advantage of hyperbolic lattices because the underlying physics of hyperbolic lattices requires more complex and intense coupling than the Euclidean one. The edge confinement requires such complex geometry despite the suppression of the field inside the bulk region.

Reply: We would like to thank the reviewer for the comment. It is true that more complex and intense couplings are required around boundaries in the hyperbolic lattice. In practice, such an effect may not always be the advantage when the simple edge connection is required. In addition, we also expect that the enhanced topological edge response could also improve the efficiency of some particular applications.

Action taken:

- In the page 5 of revised manuscript, we have added the following discussion: *“Hence, such an enhanced topological edge response may improve the efficiency of some topological devices.”.*

11. For general readers in topological wave mechanics, it may be helpful to include the discussion on the meaning of the impedance peaks, which corresponds to the transmission peaks in topological photonics or acoustics.

Reply: We would like to thank the reviewer for the comment. The impedance peaks are related to the local density of states of the corresponding quantum tight-binding model (as proved in Ref. 38).

Action taken:

- In the page 9 of the revised manuscript, we have added the following discussion to illustrate the meaning of impedance responses: *“We note that the impedance responses are related to the local density of states of the corresponding quantum tight-binding model³⁸.”.*

12. While the authors stated that the broadening in experimental results originates from the loss of elements, the theoretical results, including the loss in each element, may be included in Supplementary Materials for the completeness of the manuscript.

Reply: We would like to thank the reviewer for the comment. We have added the simulated impedance responses of hyperbolic circuits with different losses (with the effective series resistances of inductance being 20 $m\Omega$, 50 $m\Omega$, 100 $m\Omega$, and 150 $m\Omega$) in the Supplementary note 6. We find that the impedance peaks are broadening with increasing the series resistances of inductance of both hyperbolic Chern circuit and deformed hyperbolic circuit with zero modes.

Response to comments of the reviewer #2

The authors propose theoretically and realize experimentally an analog of the well-known Haldane model of a Chern insulator on a hyperbolic lattice. By numerical diagonalization in a disk geometry, the authors show the energy spectrum contains a spectral region near zero energy where eigenstates have dominant support on the sample boundary. The authors then calculate an energy-dependent real-space Chern number, and show that this Chern number exhibits a quantized plateau in roughly the same region where boundary states exist. A wave packet injected near the sample edge is shown to propagate unidirectionally and in a manner undisturbed by defects on the boundary.

The theoretical predictions are then tested experimentally on an electric circuit network. The physical system exhibits lossy (resistive) behavior that is not present in the theoretical model, but the agreement between theory and experiment is altogether good notwithstanding. In particular, the measurements demonstrate the key features expected of a Chern insulator: localized edge states within a bulk gap, chiral edge propagation, and protection against backscattering. Finally, the authors also argue that with suitable modifications, the model Hamiltonian and experimental system can support corner modes indicative of higher-order topology.

In my opinion, this is an important work that produces two key advances: a theoretical generalization of the Haldane model to hyperbolic space, and its experimental simulation using classical circuit networks. The connection between the tight-binding model and the classical circuit equations is well explained, and the experimental demonstration compelling (within the unavoidable limitation of small system sizes). The relevant literature is appropriately cited, and this work will significantly advance the burgeoning field of exotic “hyperbolic matter”. (The additional discussion of higher-order topology is somewhat less convincing, in my opinion, but I discuss this further below.)

Before I can recommend publication, I would like the authors to address the following issues/questions.

Reply: We would like to thank the reviewer for the careful review, positive evaluation and valuable suggestions of our work. In the following, we will give a detailed response to all points proposed by the reviewer.

- Although the evidence from the real-space Chern number is compelling, one would like to further check that the presence of edge states is really due to the topology of the model, as opposed to the unusual boundary/bulk ratio in the Poincaré disk, which is purely a geometrical effect. To that end, could the authors also present theoretical results in the “trivial insulator” phase, e.g., for a value of the NNN phase φ for which the real-space Chern number is zero? Ideally, one would like to see that in such a trivial insulator, there are no edge states near zero energy, and that a wave packet injected near the boundary spreads out into the bulk instead of propagating unidirectionally along the boundary. In my opinion, such a comparison between trivial and topological regimes would significantly strengthen the authors’ claims.

Reply: We would like to thank the reviewer for the comment. We calculate the eigen-spectra, the corresponding real space Chern numbers, and the dynamics of injected wave packet for systems without NNN couplings $\lambda = 0$ (in Fig. R6), and with real-valued NNN couplings $\lambda = 0.2$, $\varphi = 0$ (in Fig. R7). We can see that the real-space Chern number is trivial for both systems. And, the one-way propagations of injected wave packets are also absence.

Fig. R6. (a)-(c) The calculated eigen-spectrum, the real space Chern numbers, and the dynamics of injected wave packet for systems with the NNN coupling being $\lambda = 0$.

Fig. R7. (a)-(c) The calculated eigen-spectrum, the real space Chern numbers, and the dynamics of injected wave packet for systems with parameters being $\lambda = 0.2$ and $\varphi = 0$.

Action taken:

- In the revised manuscript, we have added the following discussion in page 5: “*And, the results of trivial hyperbolic lattice models are also provided in Supplementary Note 5 to further illustrate the difference between topological edge states and trivial edge states.*”.

- In the Supplementary note 5, we have added numerical results of the trivial hyperbolic lattice model to further illustrate the difference between topological edge states and trivial edge states.

- Where Fig. 2b-c are discussed in the text, the discussion seems to imply that the measured impedance spectrum is simply related to the local density of states (LDOS) of the corresponding quantum tight-binding model. Is there such a relation? If so, it would be nice to have a mention and/or derivation of this (e.g. in the supplementary material).

Reply: We would like to thank the reviewer for the comment. The impedance peaks are related to the local density of states of the corresponding quantum tight-binding model. Such a correspondence has been proved in Ref. 38.

Action taken:

- In the page 8 of the revised manuscript, we have added the following discussion to illustrate the meaning of impedance responses: *“We note that the impedance responses are related to the local density of states of the corresponding quantum tight-binding model³⁸.”*.

- The part of the paper I find somewhat less clear/compelling is the discussion of higher-order topological modes. First, it is not clear how the authors arrived at the model with two different coupling strengths γ_1 and γ_2 . Is this motivated by a Euclidean equivalent? This should be further discussed.

Reply: We would like to thank the reviewer for the comment. It is true that our proposed model with two different coupling strengths γ_1 and γ_2 is inspired by the C6-symmetric higher-order topological crystalline insulators in Euclidean space (Ref. 44 and Ref. 45). In particular, as for the Euclidean $\{6, 3\}$ lattice model shown in Fig. R9 (as discussed below), by introducing a pair of unbalanced intra- and intercell couplings, the zero-energy higher-order corner mode could appear (see details in the next reply).

- Related to the previous point, I am not sure if the corner modes are really a consequence of bulk topology, as opposed to trivial boundary modes arising from a cleverly engineered confining potential. It looks like the dependence of couplings on the layer index n implies the model is not translationally invariant away from the boundaries. If the authors' model were extended to the infinite lattice $L \rightarrow \infty$, would it have the full periodicity of the $\{6,4\}$ tiling?

Reply: We would like to thank the reviewer for the comment. We note that all lattice sites in the deformed hyperbolic lattice are located on the vertices of $\{6,4\}$ tiling with L approaching to infinite. And, it is true that the model is not translationally invariant with the existence of couplings depending on the layer index. Actually, previous works have demonstrated that higher-order topological corner states could exist in non-periodic systems, such as quasicrystals [Phys. Rev. Lett. 124, 036803], fractals [Phys. Rev. B 100, 155135 (2019)], and disordered systems [Phys. Rev. B 103, 085408 (2021); Phys. Rev. Lett. 125, 166801 (2020); Phys. Rev. Lett. 126, 146802 (2021)]. In the following, we demonstrate the topological properties of our proposed hyperbolic zero-energy corner states from three aspects.

Firstly, we focus on the topological phase transition induced by the unbalanced site coupling in the deformed hyperbolic lattice. As shown in Figs. R8a-R8c, we plot the calculated eigen-spectra of the deformed hyperbolic lattice with the ratio of γ_1/γ_2 being 10, 1 and 0.1, respectively. The color bar corresponds to the participation ratio (PR) of each eigen-mode. We set $L=6$, where the influence

of finite size effect could be neglected. It is clearly shown that when the coupling strength in the outermost layer is the larger one ($\gamma_1/\gamma_2=10$), the edge states are gapped around the zero energy, and there is no midgap corner state. In such a case, this bandgap is a trivial gap. The topological phase transition could appear with closing and reopening of the bandgap. It is shown that the gap is closing (around the zero-energy) with balanced values of γ_1 and γ_2 ($\gamma_1/\gamma_2=1$), corresponding to the original $\{6,4\}$ hyperbolic lattice. By further decreasing the ratio to the case with a smaller coupling strength in the outermost layer ($\gamma_1/\gamma_2=0.1$), the gap of edge states is reopened, and the midgap corner modes appear. Such a gap closing and reopening phenomenon associated with the appearance of midgap corner states is a convenient evidence for the topological phase transition.

Fig. R8. The calculated eigen-spectra of the deformed hyperbolic lattice with the ratio between γ_1 and γ_2 being 10, 1 and 0.1, respectively. Here, we set $L=6$. The color bar corresponds to the PR of each eigen-energy.

Next, we will show that the topological phase transition appearing in the deformed hyperbolic lattice is similar to the Euclidean counterpart of the C_6 -symmetric higher-order topological insulator (Ref. 44 and Ref. 45). To clearly prove this similarity, we consider the C_6 -symmetric higher-order topological insulator, as shown in Fig. R9a, where the different values of intra-cell (γ_1) and inter-cell (γ_2) couplings exist. In the following calculations, the finite C_6 -symmetric $\{6,3\}$ lattice containing 37 units is considered. As shown in Fig. R9b-R9d, we calculate the eigen-spectra of the system with the ratio between γ_1 and γ_2 being 10, 1 and 0.1, respectively. We can see that, by tuning the ratio of intra- and inter-cell couplings, the topological phase transition accompanied with the closing and reopening of bandgap happens. And, the midgap corner state could appear in the non-trivial bandgap. This phenomenon is identical to the above discussed hyperbolic topological phase transition. In this case, due to the same topological phase transition and identical symmetries (the C_6 rotation, the time reversal, and chiral symmetries), we deduce that midgap higher-order zero modes in deformed hyperbolic lattices possess similar characteristics to the filling anomaly induced 0D corner states in C_6 -symmetric higher-order topological crystalline insulators.

Fig. R9. (a) The schematic diagram of the C6-symmetric higher-order topological insulator, where the intra-cell (γ_1) and inter-cell (γ_2) couplings exist. (b)-(d) The eigen-spectra of the C6-symmetric lattice with the ratio between γ_1 and γ_2 being 10, 1 and 0.1, respectively. Here, in the calculation, the finite C6-symmetric lattice contains 37 units.

Fig. R10. (a) and (b) The calculated eigen-spectra with disorder strength being $W=0.1$ and $W=0.2$, respectively. Other parameters are set as $\gamma_1=1$, $\gamma_2=10$ and $L=6$. (c) The zero-mode distribution with $W=0.2$.

Finally, to further differentiate the topological corner modes from edge states or trivial corner states, we investigate the robustness of the midgap zero modes in the deformed hyperbolic lattice. Here, we introduce a little disorder to the onsite potential $[-W, W]$ at all bulk sites and edge sites, and keep the onsite potential of corner sites unchanged. Other parameters are set as $\gamma_1=1$, $\gamma_2=10$ and $L=6$. Figs. R10a and R10b plot the calculated eigen-spectra with W being 0.1 and 0.2, respectively. The colormap corresponds to the PR. We can see that the random onsite potential could alter the eigen-energies and localizations of trivial edge states. While, the midgap zero modes are always fixed, and the corresponding mode distribution, as shown in Fig. R10c with $W=0.2$, also keeps the same. These numerical results manifest the robustness of hyperbolic zero modes. Because, disorders should break the enhanced energy degeneracies and interference effects in the trivial hyperbolic

lattice, which makes the robust property cannot exist in the interference-induced trivial edge and corner states in hyperbolic lattices sustaining flat bands.

Action taken:

- In the revised manuscript, we have added the following discussion in page 12 to illustrate the topological properties of hyperbolic zero modes: *“While, the required translational symmetry for defining the topological index related to the higher-order topological insulator in Euclidean space⁴⁴ is absent for the hyperbolic lattice. This makes the definition of a topological invariant for the higher-order zero mode become difficult. In Supplementary Note 9, we further illustrate topological properties of our proposed hyperbolic zero-energy corner states from three aspects. Firstly, we find that the topological phase transition manifested by the closing and reopening of the zero-energy bandgap associated with the appearance of midgap corner states could appear by tuning the unbalanced site coupling in the deformed hyperbolic lattice. Moreover, we also show that the topological phase transition appearing in the deformed hyperbolic lattice is similar to the Euclidean counterpart of the C6-symmetric higher-order topological insulator^{44,54}. Finally, the robustness of the midgap zero modes in the deformed hyperbolic lattice is also proved, which cannot exist in the interference-induced trivial edge and corner states in hyperbolic lattices sustaining flat bands^{22, 32}. These features further demonstrate the topological properties of hyperbolic zero modes.”*
- In the Supplementary Note 9, we have added numerical results to demonstrate the topological properties of higher-order zero modes of deformed hyperbolic lattices.

- In Fig. 1d and/or the corresponding text, the authors should explain that the color scale corresponds to $V(\epsilon)$, which is not clear in the current version.

Reply: We would like to thank the reviewer for the kind suggestion. In the page 4 of the revised manuscript, we have added the following discussion to illustrate the colormap in Fig. 1d: *“The colormap in Fig. 1d corresponds to the quantity $V(\epsilon)$ for the localization degree at the boundary.”*

- For the real-space Chern number Eq. (2), the authors should rather cite the original reference by A. Kitaev, Ann. Phys. 321, 2 (2006).

Reply: We would like to thank the reviewer for the kind suggestion. We have cited the original paper as Ref. [35].

- In the discussion after Eq. (6) in the SM, perhaps it would be best to leave out “symmorphic”. I believe hyperbolic triangle groups do not generally have a semidirect product decomposition, though this may have to be checked.

Reply: We would like to thank the reviewer for the kind suggestion. We leave out the unsuitable word “symmorphic”.

- In Fig. S3a, the times indicated on the figure do not match those quoted in the text and the figure caption.

Reply: We would like to thank the reviewer for the comment. We have modified the incorrect times in the text and the figure caption.

- In Eqs. (9-12) in the SM, it would be preferable to use brackets (...) or [...] for matrices and vectors, instead of |...|, since the latter can be confused as a determinant. Also, Eq. (9) seems to be missing a closing parenthesis “)”.

Reply: We would like to thank the reviewer for the kind suggestion. In the revised SM, we use [...] for matrices.

- There are several typos in the text: “reach” -> “research” (p. 2); “observe above” -> “observe the above” (p. 6); “impedence” -> “impedance” several times throughout the manuscript; “eigenspectral” -> “eigenspectra” (p. 12); “remained” -> “remaining” (p. 15); “atom limit” -> “atomic limit” (p. 9 of SM). Also, the phrase “with extremely fewer trivial regions” at the end of the abstract is rather unclear. I would suggest something like “which maximize the topological edge response”.

Reply: We would like to thank the reviewer for the kind suggestion. We have modified these grammatical and spelling errors in the manuscript.

REVIEWERS' COMMENTS

Reviewer #1 (Remarks to the Author):

In this revised manuscript, the authors provided very careful and complete responses to the comments and suggestions raised by the reviewers. Most importantly, the authors successfully demonstrated the topological features of the zero modes, leading to the consistency of the entire manuscript. The other responses on comprehensible explanations of the concepts and revised notations and parameters also improved the completeness of the manuscript. The reviewer thanks the authors' efforts during the revision and happily suggests the acceptance of the manuscript.

Reviewer #2 (Remarks to the Author):

The authors have answered my questions satisfactorily, and the new theoretical results they have added to the paper (in particular, results for a topologically trivial Hamiltonian, and a better explanation of the C_6 -protected higher-order zero modes) have significantly improved the manuscript. In my opinion, the manuscript is now suitable for publication in Nat. Commun.

Response Letter to Reviewers

(NCOMMS-NCOMMS-22-06101A)

Response to comments of the reviewer #1

In this revised manuscript, the authors provided very careful and complete responses to the comments and suggestions raised by the reviewers. Most importantly, the authors successfully demonstrated the topological features of the zero modes, leading to the consistency of the entire manuscript. The other responses on comprehensible explanations of the concepts and revised notations and parameters also improved the completeness of the manuscript. The reviewer thanks the authors' efforts during the revision and happily suggests the acceptance of the manuscript.

Reply: We would like to thank the reviewer for the acceptance of our revised manuscript.

Response to comments of the reviewer #2

The authors have answered my questions satisfactorily, and the new theoretical results they have added to the paper (in particular, results for a topologically trivial Hamiltonian, and a better explanation of the C_6 -protected higher-order zero modes) have significantly improved the manuscript. In my opinion, the manuscript is now suitable for publication in Nat. Commun.

Reply: We would like to thank the reviewer for the acceptance of our revised manuscript.